# Monitoring Universal Health Coverage reforms in primary health care facilities: Creating a framework, selecting and field-testing indicators in Kerala, India

**Devaki Nambiar**[1,2,3]**, Hari Sankar D.** [1]***, Jyotsna Negi**[4]**, Arun Nair**[5]**, Rajeev Sadanandan**[6]

**1** The George Institute for Global Health India, New Delhi, India, **2** Faculty of Medicine, University of New South Wales, Sydney, Australia, **3** Prasanna School of Public Health, Manipal Academy of Higher Education, Karnataka, India, **4** Independent Consultant, Baltimore, MD, United States of America, **5** ACCESS Health International Inc, New Delhi, India, **6** Health Systems Transformation Platform, New Delhi, India

* hsankar@georgeinstitute.org.in, vaidyarsankar@gmail.com

## Abstract

In line with the Sustainable Development Goals (SDGs) and the target for achieving Universal Health Coverage (UHC), state level initiatives to promote health with "no-one left behind" are underway in India. In Kerala, reforms under the flagship Aardram mission include upgradation of Primary Health Centres (PHCs) to Family Health Centres (FHCs, similar to the national model of health and wellness centres (HWCs)), with the proactive provision of a package of primary care services for the population in an administrative area. We report on a component of Aardram's monitoring and evaluation framework for primary health care, where tracer input, output, and outcome indicators were selected using a modified Delphi process and field tested. A conceptual framework and indicator inventory were developed drawing upon literature review and stakeholder consultations, followed by mapping of manual registers currently used in PHCs to identify sources of data and processes of monitoring. The indicator inventory was reduced to a list using a modified Delphi method, followed by facility-level field testing across three districts. The modified Delphi comprised 25 participants in two rounds, who brought the list down to 23 approved and 12 recommended indicators. Three types of challenges in monitoring indicators were identified: appropriateness of indicators relative to local use, lack of clarity or procedural differences among those doing the reporting, and validity of data. Further field-testing of indicators, as well as the revision or removal of some may be required to support ongoing health systems reform, learning, monitoring and evaluation.

## Introduction

The Sustainable Development Goals (SDGs), which currently guide national agendas for health, have set a target for achieving Universal Health Coverage (UHC), including financial risk protection, access to quality essential health care services and access to safe, effective,

**Data Availability Statement:** All relevant data are within the paper and Supporting Information files.

**Funding:** We wish to indicate that this work was supported by the Wellcome Trust/DBT India Alliance Fellowship (https://www.indiaalliance.org) Grant number IA/CPHI/16/1/502653) awarded to Dr Devaki Nambiar. The funder had no role in study design, data collection and analysis, decision to publish, or preparation of the manuscript. The funder provided support in the form of salaries and research materials and fieldwork support for authors DN, HS and JN but did not have any additional role in the study design, data collection and analysis, decision to publish, or preparation of the manuscript. The specific roles of these authors are articulated in the 'author contributions' section.

**Competing interests:** We further wish to clarify that AN is employed by ACCESS Health International (https://accessh.org/ - not ACCESS Health. ACCESS Health International is a registered not for profit organisation and is not a commercial company. We are in full adherence of PLoS One policies on sharing data and materials. I wish to confirm that all authors have no competing interests to declare.

**Abbreviations:** ANC, Ante-Natal Care; CD, Communicable Disease; CPHC, Comprehensive Primary Health Care; FHC, Family Health Centre; GO, Government Order; HWC, Health & Wellness Centre; LMIC, Low and Middle-Income Countries; M&E, Monitoring and Evaluation; NCD, Non-Communicable Disease; PHC, Primary Health Centre; PROGRESS, Place of Residence; Race/ethnicity/culture/language; Occupation; Gender/sex; Religion; Education; Socioeconomic status; Social capital; RMNCAH, Reproductive, Maternal, Newborn, Child and Adolescent Health; SDG, Sustainable Development Goal; SHSRCK, State Health Systems Resource Centre Kerala; UHC, Universal Health Coverage; WHO, World Health Organisation.

affordable and quality essential medicines and vaccines for all as goal 3.8 [1]. Evidence from Low and Middle-Income Countries (LMIC) has supported the effectiveness of improved and better resourced primary health care services in achieving UHC and improving health outcomes [2], and reducing inequalities in population health outcomes [3]. This is underscored in the 2018 Astana Declaration on Primary Health Care as well [4].

India's 2017 National Health Policy foregrounds equity in calling for free access to primary health care, improved access and affordability of secondary and tertiary level care, and reduction of out of pocket expenditure [5]. In the southern Indian state of Kerala, UHC-relevant reforms have been in place for over a decade. In 2016, following a detailed process of setting up state level Sustainable Development Goals (SDGs), a health reform measure, the Aardram Mission was announced, which included the transformation of existing Primary Health Centres (PHCs) into Family Health Centres (FHCs). Kerala's FHCs correspond to the vision of comprehensive primary health care through conversion of subcentres to Health and Wellness centres introduced by the national government the following year [6,7].

Aardram and the FHC program were designed bearing in mind that while Kerala's health system has made great gains in addressing maternal and child health outcomes [8], it has also witnessed an epidemiological transition. The state faces a substantial burden of diabetes and cancers [9,10], must reckon with unregulated privatization and catastrophic health spending [11,12], as well as recalcitrant challenges in controlling emerging and vaccine preventable diseases [13,14]. Lack of faith in the primary care system has led to PHCs being bypassed in favour of tertiary hospitals, leading to higher expenditure for patients and inefficient use of health resources.

Tracking Aardram's various reforms requires monitoring and evaluation (M&E), which is also seen as a cornerstone of UHC, mindful of the variations in geography, resources, morbidity patterns, and other contextual factors. Availability of routine and disaggregated health and health systems data is vital in this process. Critically, a UHC framework requires attention to the notion of coverage- of those eligible for a service, how many are receiving it and who is getting left behind? The challenge before the state was to adapt its monitoring framework–which largely relied on reporting progress against targets—to assess coverage, i.e. progress against a denominator of the eligible population. Further, while FHC reforms were extensive and aimed to be comprehensive, a need existed for a shortlist of tracer indicators to track progress in reforms against state SDGs, as well as relevant national and global benchmarks. Providing real-time feedback to local and state level managers was also envisioned, to increase efficiency and transparency of the reform process. These considerations led the team to collaborate on the development and testing of UHC-relevant coverage indicators for the state to assess the progress of FHC reforms.

## Methods

This study employed a modified Delphi method; which has been widely used to develop consensus in primary care monitoring indicator development and questions related to health policy [15–17]. The Delphi method relies on domain knowledge and experience of an expert panel whose members rank a structured questionnaire and consensus is reached based on collated ranking responses. In our case, the Delphi method was modified to help to arrive at a consensus among experts, whose responses to the indicator list were ranked, aggregated, revised and then sent back to a larger group of experts to develop a robust list of indicators. Institutional Ethics Committee approval was obtained from the George Institute for Global Health (Project Numbers 08/2017 and 05/2019). The following six steps were involved in the process:

First, we developed a conceptual framework drawing from global and local references and field visits. We used as a reference global monitoring tools of the Joint Learning Network for UHC as well as the SDG 3 UHC target tracer indicators, and the WHO's health systems impact framework [18–20]. In addition, we compiled and reviewed all relevant Government Orders (GO) and policy documents, as well as reports of the working groups related to FHCs.

Policy document review was triangulated in the second step with field visits to primary health care facilities in three geographically distinct districts where we carried out discussions with key stakeholders (n = 12). Based on this, a conceptual framework and an inventory of 812 indicators were constructed drawing from state, national, and global inventories.

The third step involved mapping the data sources used for monitoring at the facility, district and state level. The data source list was created by visiting health centres from three districts: nearly 89 unique registers in FHCs were identified and given a unique data source number. Where possible, we noted which dimension of inequality using PROGRESS Plus (i.e. place of residence, race/ethnicity, occupation, gender, religion, education, social capital, socioeconomic status, plus age, disability) was included by data source [21].

As a fourth step, based on multiple field visits, expert consultations, a desk review of existing data sources and process, and a team-based effort linking global/national primary health monitoring frameworks to Kerala priorities and processes, we arrived at a list of 38 indicators. Our construction of the health equity indicators was largely informed by a list of sixteen tracer indicators selected by WHO's Global Monitoring Report [20]. The shortlisted indicators were classified into domains, sub-domains, lowest level of analysis, periodicity, and availability of data related to dimensions of inequality. For ease in understanding the method to arrive at the final value of the indicators, numerator and denominator definitions of each indicator were indicated, along with potential sources. In this process, we noted that population-based surveys (eg. the National Family Health Survey and National Sample Survey Organization Consumption Expenditure and Health Surveys) are highly reliable at the state and national level but are not powered to provide estimates at the PHC (30,000 population) level. Further, their periodicity is inadequate to guide course correction at the PHC level, as they are conducted after every three or five years. Therefore, alternate sources used at the facility level, like the antenatal care register, Reproductive Child Health (RCH) Register/portal, and Non-Communicable Disease (NCD) Surveillance Register were identified.

The fifth step was a modified Delphi, designed to ensure representation of a range of expert views and experiences: including frontline health workers, domain experts, senior bureaucrats and implementers in two rounds. Eligibility criteria for participation were: experience of more than five years in relevant domain; firsthand experience with primary care in Kerala; and past or present formal role in design and/or delivery of the FHC program.

In the first round, six participants were invited to a group meeting where team members clarified the selection and definition of the initial shortlist of 38 indicators and the modified Delphi process. A ranking tool was given to participants in this stage. A rank of 1 denoted that the indicator had the highest possible priority (i.e. the indicator is important and/or should be monitored on priority) while a ranking of 5 indicated that this was the lowest priority indicator (i.e. the indicator is not as important and/or can be prioritised for monitoring later).

Participants also had an option of adding or removing indicators under each domain area–in which case justification was to be provided, including what the data source for the indicator could be. Participants were provided with an inventory of 812 proxy indicators created by the team at the initial stage. Data was collected in hard copy: participants completed their ranking using a pen or pencil. All responses from round 1 were reviewed, compiled and revised to be used for Round 2. A total of 22 indicators were added (after removing duplicates) and 15

indicators were removed by consensus. A revised list of 31 indicators was constructed to be used at Round 2.

In the second round, indicators were classified under five subdomains and experts from the state on 1) communicable diseases (CD), 2) governance and financial protection, 3) non-communicable diseases (NCDs), injuries and palliation, 4) reproductive, maternal, neonatal, child and adolescent health (RMNCAH), and 5) service delivery were identified (n = 25). The ranking tool was sent in soft copy to the experts. Ranking instructions and a ranking survey tool contained detailed instructions about the process. The inventory of global indicators (n = 812) was also sent to experts as reference material. Participants had the option of recommending additional indicators. To reduce bias in ranking, the tool consisted of specific information on consideration of how to rank an indicator (S1 File). Experts were encouraged to discuss with their colleagues in the respective domain about the relevance of individual indicators in a comprehensive monitoring framework. A team member visited the experts individually after sending them ranking tool to clarify and doubts in ranking procedure and reiterated the principles to rank them.

Following standard convention and the procedure undertaken in prior ranking exercises [22], mean and median priority scores were calculated for each indicator by creating decision rules based on the distribution of ranks. For the final indicator list, indicators that received a median rank of 1 were included (Table 1). Further, Indicators that received a median rank greater than 1 were included only if the indicator was ranked higher (ie, the value was lower than) than 2.5. The shortlisted indicators were finally presented to senior health administrators at state level for final vetting and approval.

**Table 1. List of monitoring indicators for primary care arrived at following modified Delphi and consultation.**

| Sr No | Domain | Sub Domain | Periodicity | Indicator Name | Numerator | Denominator | Data Source | Median Rank | Mean Rank | Availability in prescribed form at FHC/PHC |
|---|---|---|---|---|---|---|---|---|---|---|
| AP1 | Outcome | RMNCH+A | Annual | Proportion of pregnant women who received all recommended types of antenatal care (ANC) for the last live birth within a stipulated time period | For the last live birth, Number of mothers who received four or more antenatal checks, received at least one tetanus toxoid injection, and took iron and folic acid tablets or syrup for 100 days or more | Number of women with a live birth in a given time period | Reproductive Child Health (RCH) Register/ portal | 1 | 2.4 | Partially available Numerator or denominator available |
| AP2 | Outcome | RMNCH+A | Annual | Full immunisation coverage rate | For vaccines in the infant immunization schedule, this would be the number of children aged 12–13 months who received the specified vaccinations before their first birthday | Total number of infants surviving to age one | RCH Register/portal | 1. | 1.5 | Partially available Numerator or denominator available |
| AP3 | Outcome | RMNCH+A | Monthly | Incidence of low birth weight among newborns | Number of live-born neonates with weight less than 2500 g at birth | Number of live births | RCH Register/portal | 2 | 2.1. | Available, not reliable |
| AP4 | Output | Communicable Diseases | Monthly | Breteau index Score (calculated per 100 households) | Number of containers in which larvae are found (positive containers) per 100 households inspected | Nil | Vector Surveillance Register | 2 | 2.1 | Available not reliable |
| AP5 | Outcome | Communicable Diseases | Monthly | Incidence of Acute Diarrhoeal Disease (ADD) among children under five | Number of New ADD cases within the stipulated time period | Population at risk (number of children under five years -existing ADD cases) | Out Patient (OP) register, S form, Integrated Disease Surveillance Programme (IDSP) Register, Family Health Register | 2 | 2.1 | Partially available Numerator or denominator available |
| AP6 | Outcome | Communicable Diseases | Annual | Tuberculosis (TB) treatment completeness coverage rate | Number of new and relapse TB cases that were notified and completed treatment within the stipulated time period | Number of TB cases in the same period | TB Register, Nikshay portal | 2 | 2.3 | Available |

(*Continued*)

**Table 1.** (Continued)

| Sr No | Domain | Sub Domain | Periodicity | Indicator Name | Numerator | Denominator | Data Source | Median Rank | Mean Rank | Availability in prescribed form at FHC/PHC |
|---|---|---|---|---|---|---|---|---|---|---|
| AP7 | Output | NCDIs and Palliative care | Monthly | Proportion of eligible adults (aged 30 years or above) who had blood pressure screening within a stipulated time period | Number of eligible adults whose blood pressure was measured within a stipulated time period | Number of adults aged 30 or over | Non Communicable Disease (NCD) Surveillance Register | 1 | 2.1 | Partially available Numerator or denominator available |
| AP8 | Output | NCDIs and Palliative care | Monthly | Proportion of eligible adults (aged 30 years or above) who had blood glucose screening within a stipulated time period | Number of eligible adults whose blood glucose was measured within a stipulated time period | Number of adults aged 30 or over | NCD Surveillance Register | 1 | 2.1 | Partially available Numerator or denominator available |
| AP9 | Outcome | NCDIs and Palliative care | Monthly | Proportion of those screened at PHC/FHC diagnosed with depression within a stipulated time period | Number of persons diagnosed with depression within a stipulated time period | Total population screened for depression | Aswaas Clinic Register | 2 | 2.1 | Available |
| AP10 | Output | NCDIs and Palliative care | Monthly | Number of patients receiving palliative care services within a stipulated time period | Number of patients receiving palliative care services within a stipulated time period | Nil | Palliative care register, Primary data collection | 2 | 2.4 | Available |
| AP11 | Input | Governance, Stewardship and Financing | Annual | Per capita current Primary Health Centre (PHC)/Family Health Centre (FHC) expenditure | Total current expenditure on health by PHC/FHC | Total population | State Budget sheets/ Budget sheet of the PHC | * | * | Not a facility level indicator |
| AP12 | Input | Governance, Stewardship and Financing | Annual | Proportion of Local Self-Government Institutions (LSGI) funds spent for health within a stipulated time period on- a) special populations (slum dwellers, tribal groups, migrant labourers) and b) social determinants (drinking water, nutrition, sanitation, overall convergence) | Total LSGI funds spent on a) special populations (slum dwellers, tribal groups, migrant labourers) and b) social determinants (drinking water, nutrition, sanitation, overall convergence) within a stipulated time period | Total LSGI funds released for health in the same period | Sankhya portal | 1 | 1.7 | Partly available, disaggregation not always available |
| AP13 | Input | Governance, Stewardship and Financing | Monthly | Proportion of FHCs that submitted core reports to the district within a stipulated time period | Number of FHC that submit core reports to the district within stipulated time period | Total FHC | Health Management Information System (HMIS) | 2 | 2.4 | Not a facility level indicator |
| AP14 | Input | Governance, Stewardship and Financing | Monthly | Proportion of registered clinical establishments reporting to IDSP within a stipulated time period | Number of registered clinical establishments reporting to IDSP within a stipulated time period | Total number of registered clinical establishments | IDSP register | 1. | 2.4 | Not a facility level indicator |
| AP15 | Output | Service capacity and access | Monthly | Proportion of patients referred from FHC to higher levels within a stipulated time period | Number of cases referred from FHC to higher level within a stipulated time period | Total number of cases seen at facility during the measurement period | Referral Register | 1.5 | 2.1 | Available |
| AP16 | Input | Service capacity and access | Annual | Proportion of Vacant healthcare provider positions (regular + contractual) in PHC/FHC within a stipulated time period for following a. Auxiliary Nurse Mid-wife (ANM) at sub-centers (SCs) b. Junior Health Inspectors at PHCs c. Staff nurse (SN) at Primary Health Centers (PHCs) d. Medical officers (MOs) at PHCs | Number of vacant positions at PHCs/FHCs within a stipulated time period | Total sanctioned healthcare provider positions for following cadres (separately for each cadre) during a specific year: | Staff Position Reports (Digital) | 1 | 2.1 | Available |
| AP17 | Output | Service capacity and access | Monthly | Proportion of all out-patients receiving pre-check service by Staff Nurse within a stipulated time period | Number of patients receiving pre-check service by Staff Nurse within a stipulated time period | Total number of outpatient's visits | Pre-check register, OP registration register | 1 | 1.6 | Available |
| AP18 | Input | Service capacity and access | Monthly | Proportion of health facilities with Kerala-recommended core list of essential medicines available within a stipulated time period | Number of health facilities with Kerala- recommended core list of essential medicines available in stock within a stipulated time period | Total health facilities | Drug Distribution Management System | 1 | 1.1 | Not a facility level indicator |
| AP19 | Input | Service capacity and access | Monthly | Proportion of facilities providing extended service hours | Number of facilities providing extended service hours | Total health facilities | Staff Position Reports (Digital) | 2 | 1.8 | Not a facility level indicator |
| AP20 | Output | Service capacity and access | Monthly | Daily caseload per doctor | Number of outpatient visits recorded in outpatient records in the health facility in the prior month | Average daily attendance of doctors across all days the facility was open. | OP registration register, Attendance register | * | * | Available |
| AP21 | Output | Service capacity and access | Monthly | Diagnostics tests per lab technician | Number of diagnostic tests at the health facility within a stipulated time period | Total number of lab technician in a health facility (Regular + contractual) | Lab Register | * | * | Available |

(Continued)

**Table 1.** (Continued)

| Sr No | Domain | Sub Domain | Periodicity | Indicator Name | Numerator | Denominator | Data Source | Median Rank | Mean Rank | Availability in prescribed form at FHC/PHC |
|---|---|---|---|---|---|---|---|---|---|---|
| AP22 | Outcome | NCDIs and Palliative care | Monthly | Number of eligible adults (aged 30 years or above) put on hypertension management whose Blood Pressure was within target range after a year of treatment. | Number of eligible adults (aged 30 years or above) put on HTN management whose BP was within target range after a year of treatment. | NA | Non Communicable Disease (NCD) Clinic Register | * | * | Not piloted, to be included in primary survey |
| AP23 | Outcome | NCDIs and Palliative care | Monthly | Number of eligible adults (aged 30 years or above) put on Diabetes Mellitus management whose Fasting Plasma Glucose (FPG) level was within target range after a year of treatment | Number of eligible adults (aged 30 years or above) put on DM management whose FPG level was within target range after a year of treatment | NA | Non Communicable Disease (NCD) Clinic Register | * | * | Not piloted, to be included in primary survey |
| R1 | Outcome | RMNCH+A | Annual | Proportion of pregnant women tested positive for Gestational Diabetes Mellitus (GDM) within a stipulated time period | Number of pregnant women tested positive for GDM within a stipulated time period | Number of registered pregnant women | RCH Register/portal | These indicators were recommended by two or more experts. Therefore, they were added to the list as recommended indicators. | | Partially available Numerator or denominator available |
| R2 | Outcome | RMNCH+A | Monthly | Proportion of adolescent girls aged 11–19 years with anaemia within a stipulated time period | Number of adolescent girls aged 11–19 years with anaemia within a stipulated time period | Number of adolescent girls aged 11–19 years screened for anaemia | School Health Register | | | Partly available, lacks required disaggregation |
| R3 | Outcome | RMNCH+A | Annual | Proportion of children aged 0–23 months who were born at least 24 months after the previous surviving child | Number of children aged 0–23 months who are at least 24 months younger than the previous surviving sibling | Total number of children aged 0–23 months with a next older sibling | RCH Register/portal | | | Available |
| R4 | Outcome | Communicable Diseases | Annual | Number of communicable disease outbreaks within a stipulated time period* | Number of communicable disease outbreaks within a stipulated time period | | IDSP register/ Primary data collection | | | Available not reliable |
| R5 | Outcome | Communicable Diseases | Monthly | Proportion of cases of fever reported as malaria within a stipulated time period | Number of fever cases reported to be malaria within a stipulated time period | Number of fever cases | HMIS | | | Partially available Numerator or denominator available |
| R6 | Output | Communicable Diseases | Monthly | Proportion of people with asthma and COPD care coverage within a stipulated time period* | Number of people diagnosed with asthma under treatment + Number of people diagnosed with COPD under treatment within a stipulated time period | Number of people enrolled in Swaas clinic | Swaas Register | | | Available not reliable |
| R7 | Outcome | RMNCH+A | Annual | Proportion of children under 5 years who are stunted (moderate and severe) within a stipulated time period | Number of children aged 0–59 months who are stunted within a stipulated time period | Total number of children aged 0–59 months | Anganwadi Weight Monitoring Chart | | | Available not reliable |
| R8 | Outcome | Communicable Diseases | Annual | Proportion of children under 15 among all new leprosy cases detected within a stipulated time period | Number of children aged under 15 years detected leprosy within a stipulated time period | Number of new cases detected for leprosy during the same year. | Leprosy Register | | | Partly available, lacks required disaggregation |
| R9 | Output | Service capacity and access | Monthly | Proportion of total outpatients seen in the evening within a stipulated time period | Number of evening outpatient visits recorded in outpatient records in the health facility within a stipulated time period | Total outpatient load (evening + morning) | OP registration register | | | Available |
| R10 | Input | Service capacity and access | Annual | Proportion of facilities in which > = 50% of providers report receiving formal training related to their work within a stipulated time period | Number of facilities in which > = 50% of providers report receiving pre-service or in-service training related to their work within a stipulated time period | Number of facilities surveyed | Primary data collection | | | Not a facility level indicator |
| R11 | Input | Service capacity and access | Monthly | Average time between registration and time of physician initial assessment. | Average time between registration or triage and time of physician initial assessment. | Nil | Primary data collection | | | Not piloted, included in primary survey |
| R12 | Input | Service capacity and access | Monthly | Proportion of institutions like old age homes, factories visited by a staff nurse within a stipulated time period | Total number of institutions visited by a staff nurse like old age homes, factories within a stipulated time period | Total number of institutions identified for institutional care like old age homes, factories in FHC catchment area | Primary data collection | | | Available |

*These indicators were included specifically on instruction from a senior health administrator to track the progress of the new program and were not ranked by experts.

In the sixth step, the indicator list was field-tested in three purposively chosen FHCs in Kerala to have geographic spread across the state and reflect a range of circumstances (i.e. catering to peri-urban, tribal and rural areas). We explained the definitions and logic of all the

indicators to the staff and received their inputs on refining their reporting and calculation. Based on this, a structured data collection template for collecting the data from the previous year (2017–18) corresponding to the indicators was prepared. This format was emailed to these facilities. The data collected from the facilities were analyzed and feedback was given to the facilities.

## Results

The conceptual model for this study was developed drawing from multiple sources [6,23–27] (Fig 1). It comprised principles, inputs, outputs, and outcomes, while also acknowledging intersectoral linkages and community participation as envisioned in the Aardram Mission.

A total of 31 participants were invited to participate in two Delphi rounds, out of whom 25 responded, a response rate of 80% (Fig 2). Four out of six participants ranked in the first round and 21 out of 25 participants ranked in the second round. Nearly half of the experts (48%) who participated in two rounds of the exercise were above fifty years of age and almost two-thirds were male (64%) (Table 2). Most experts had domain experience of more than ten years (76%) and nearly half of them (48%) were currently serving as state level specialist consultants or program officers for public health programs. Participants also included those working in a grassroots implementation like field-level health workers, and primary care doctors–almost all experts had experience delivering primary health care at some stage of their careers.

After ranking responses and discussions in the first round of the modified Delphi process, the list was reduced to 31 indicators. Additional indicators (N = 56) were suggested. Duplicates were removed and in the second round, additional ranking took place in the course of which 12 indicators recommended by two or more participants were included in the final list as

| PRINCIPLES | | |
|---|---|---|
| Universal, family-basis, equity and non-discrimination, comprehensiveness, financial risk protection, quality, rationality, portability and continuity of care, protection of patient rights, community participation, accountability and responsiveness. | | |
| **INPUTS** | **OUTPUTS** | **OUTCOMES** |
| Stewardship, Organization & Quality Improvement<br>Health information systems<br>Human resources<br>Financing<br>Essential medicines, diagnostics and devices | Field based service delivery<br>Facility based service delivery<br>Target linked coverage (may be embedded above) | Health Outcomes (Mortality, Morbidity, Risk Factors)<br>Financial risk protection<br>Responsiveness<br>Efficiency |
| **INTERSECTORAL LINKAGES** | | **COMMUNITY PARTICIPATION** |
| Linkages to various departments (police, social security mission, social justice, priority schemes) | | Arogyasenas<br><br>Social audits |

**Fig 1. Conceptual framework for Family Health Centre (FHC) monitoring and evaluation and equity analysis.**

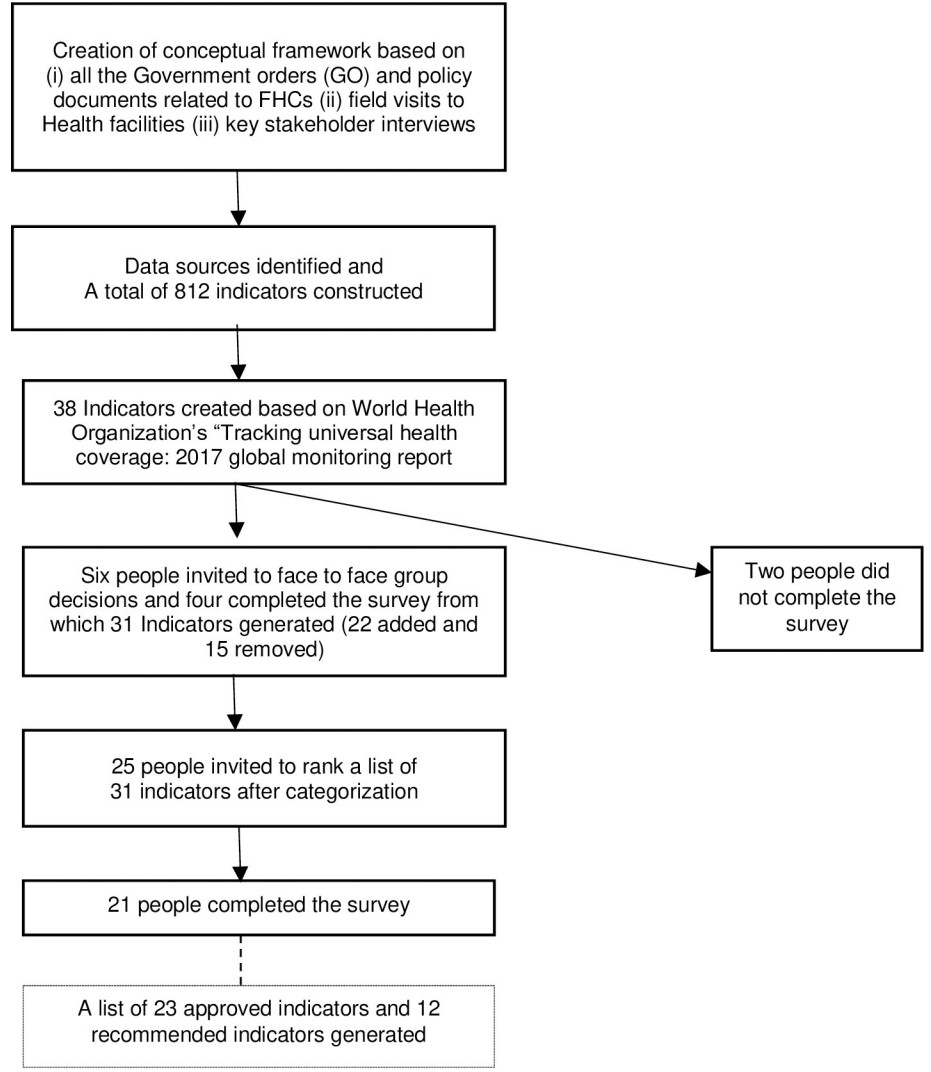

**Fig 2. Flow chart of the modified Delphi process.** Source: Authors.

recommended (by experts, Fig 2). The list was finalised with a senior health official, resulting in an indicator list of 23 approved indicators and 12 recommended indicators (Table 1).

The 23 approved indicators were a mix of input indicators (n = 7), output indicators (n = 8), and outcome indicators (n = 8). There were 3 indicators for tracking communicable diseases, 6 for NCD, injuries and palliation, 3 for RMNCAH, 7 for service delivery and 4 related to governance, stewardship, and financial protection. Most of these indicators (n = 15) were to be measured at the FHC level, some (n = 5) of them were to be measured at the district level. Further, 17 of these indicators were to be measured monthly and six of them were designed to be measured annually.

We undertook a facility-based field test of relevant indicators, ie. those that would be available/gathered at the facility. Data for eight indicators were readily available in facilities (Table 1). For the remaining, field testing revealed three types of challenges (see Fig 3); problems with the operationalisation of our pre-defined indicators; problems with the process(es) of monitoring/reporting indicators; and lack of gold standard or issues of validity/reliability of indicators.

**Table 2. Basic profile of experts who participated in two rounds of modified Delphi method to develop monitoring indicators for primary care in Kerala.**

| | Number (percentage) of participants) (N = 25) |
|---|---|
| **Age** | |
| **30–40 years** | 5 (20) |
| **40–50 years** | 8(32) |
| **50–60 years** | 7(28) |
| **60 years and above** | 5(20) |
| **Gender** | |
| **Male** | 16(64) |
| **Female** | 9(36) |
| **Domain of Primary Expertise** | |
| **Non-Communicable Diseases, Injury and Palliative care** | 8(32) |
| **Reproductive, Maternal, Neonatal, Child, and Adolescent Health** | 2(8) |
| **Communicable Diseases** | 4(16) |
| **Health Financing** | 2(8) |
| **Service capacity and delivery** | 9(36)() |
| **Years of domain experience** | |
| **5–10 years** | 6(24) |
| **10–20 years** | 10(40) |
| **20 years and above** | 9(36) |
| **Job profile** | |
| **Field health worker** | 3(12) |
| **Primary Care doctor** | 2(8) |
| **Program officer/Program consultant** | 10(40) |
| **Academic** | 4(16) |
| **Senior state health administrator** | 4(16) |
| **District health administrator** | 2(8) |
| **Institutional Affiliation** | |
| **Department of Health services** | 12(48) |
| **National Health Mission** | 5(20) |
| **Department of Medical Education** | 2(8) |
| **Other (includes academic institutions, private facilities, multilateral institutions and state agencies)** | 6(24) |

With regard to the first challenge, in the case of two indicators (full Antenatal Care (ANC) coverage and full immunisation coverage) numerators were available but both were being reviewed at the facility level using predetermined annual targets as denominators.

In the case of indicators related to newer aspects of Aardram, like depression screening at FHCs, we found that the denominator (number of patients who visited the clinic) and numerators (number of patients detected with depression) were almost identical. It was clarified that people who came to the clinic first visited the doctor and then were screened for depression if referred by the doctor. This is not how screening is envisioned in the programme, but it is the practice and must be considered when interpreting results. In other cases, data was being filled in by contractual staff, who had not been trained and while they may have had required clinical competencies, they likely lacked training in guidelines of specific programmes and reporting requirements.

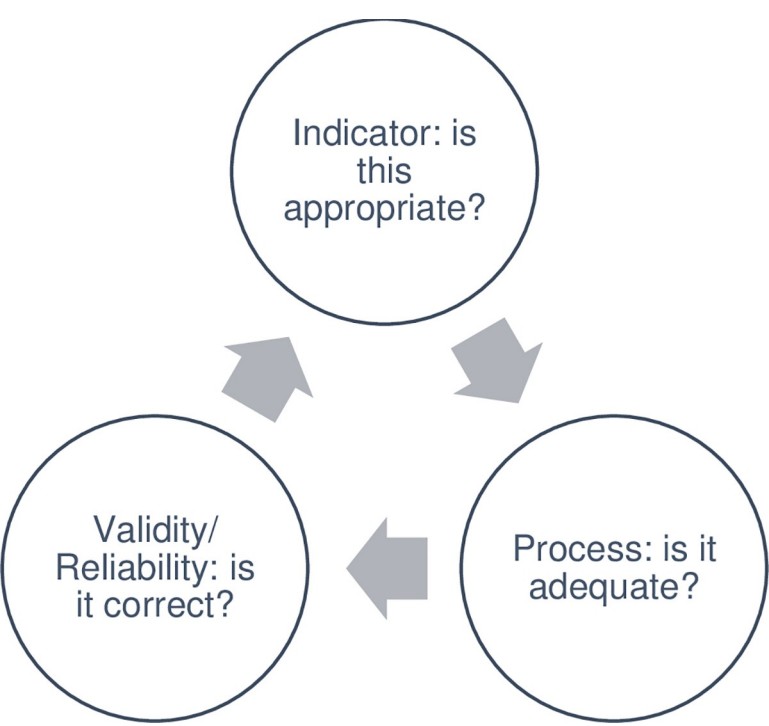

**Fig 3. Issues raised after field testing Delphi-approved indicator list.** Source: Authors.

Indicators on low birth weight and the Breteau vector index had data reliability issues due to varied methods of data collection and knowledge level of field staff. Methods used in the field to generate these data were not standardized and varied widely; as such they could not be adjudged to be valid.

From our field testing, we noted that it may be required to drop indicators altogether and replace them with more appropriate, field-validated ones. Such recommendations and feedback were duly conveyed to the Aardram leadership and will continue to be tracked as the FHC program advances. Currently, apart from primary triangulation of these indicators through a primary household survey, formats are being created to continue to field test indicators over a longer period, with training on entry, analysis and use of the data proposed at the facility, program, and state-level supervisory levels.

It may eventually be feasibly to develop summary measures or indices for PHC reform in the state, allowing facility and district comparisons and benchmarking and monitoring of inequalities. We found that disaggregated data was available for many indicators by sex. However, data on other dimensions of inequality was not available (even as plans to create linking identifiers are underway). Some indicators (like adolescent anaemia) at some facilities are calculated for tribal populations separately- an important equity-sensitive enhancement that is not currently reflected in the larger M&E structure. Requests were made at some facilities to include location-specific disease burdens (like Kyasanur Forest Disease prevalent in tribal areas of Wayanad district) that disproportionately affect tribal and remote populations. The team indicated that such bespoke indicators could be monitored in individual facilities by their own prerogative for periodic updating, acknowledging that doing this would also require time and resources. However, from an equity lens and responsiveness to health needs of the local community, the value of such customization is immense.

## Discussion

In this Delphi study, an expert panel consisting of key health system actors ranked a set of indicators for routine measurement of primary health service coverage and system performance. The intention was for this shortlist of indicators to be used at the facility level to assess the progress of UHC-relevant reforms in the state of Kerala. Persons working in the health system are more aware of the utility of the indicators to inform them of the performance of the system and feasibility of gathering the data from within the system. For this reason, we sought to involve them closely in the selection and development of indicators. The main challenges in tracking UHC found in this exercise in Kerala were reliability of data sources, the absence of disaggregated data to determine coverage inequities, and challenges in the measurement of effective coverage, which includes the impact of services on the health of people rather than mere service utilisation [28].

In light of ongoing reforms, moreover, the attempt was made to embed processes of concurrent monitoring so as to provide continuous lessons and course correction in programme rollout. A key challenge we have faced is related to indicators–we have found that global indicators tend to most easily afford global comparisons but are often not relevant at the local, programmatic level because they do not speak to key features of implementation. Even well-established indicators like ANC care coverage over live births or facilities with essential medicines are important to national and international bodies for interstate comparisons but can be of little relevance to local managers who must monitor against targets with high periodicity (i.e. monthly monitoring of ANC) and dynamic changes (Kerala facilities have the discretion to define essential medicines based on local needs). Local managers demand data that indicates how well the health needs of the population assigned to them are being addressed. Critically, perceived utility by the persons who collect the data has the potential to influence the quality of the data collected. This, in turn, will impact comparisons and benchmarking at higher levels: as a recent burden of disease study has pointed out "countries require open-source, locally operable, transparent, and believable data paired with simple, transparent and reproducible tools to track progress towards the 2030 UN Sustainable Development Goals" [29].

More broadly, It is well established that institutions with varying purposes and scope–global versus local, but also institutions that deliver services and those that make public health decisions–may have different logics matching their institutional goals, shaped by varying legacies, stakeholders and histories [30]. In each case, custodians and users of data shift and commensurably, we have found, there is a "wiggle" in the definitions and operationalisations of indicators (for instance in determining depression screening coverage or the Breteau index value for an area). On the other hand, in light of these legacies and institutional complexities, "the single window of truth" has proven difficult to achieve as existing systems fight hard to retain their existence. . . .this 'single version' requires protocols for data comparison and error management, and audit trails for tracking what changes are made to the data, where, and by whom" [30]. Jordans and colleagues [31] in a Delphi study conducted to develop routine monitoring indicators for mental health in Low and Middle Income Countries note that HMIS systems across the globe are burdened by exhaustive data collection, which invariably raises questions on the quality of the collected data. In the present study, the intention was to select a shortlist of tracer indicators, through a process of attrition at each step of the Delphi, yet when we visited facilities, there was a demand for bespoke indicators to be added to the list. Any monitoring process should be nimble so that it is not placing an undue burden on the system while also ensuring that emerging burdens and concerns of local staff are visible and addressed. This process has therefore derived from and shall continue to rely on routine data for indicator generation: as aforementioned, close collaboration and integration with the state's comprehensive

e-health initiative is envisioned, with validation from survey data, a recommended good practice [32,33]. However, our list, unlike many other inventories, spanned a wider berth of domain areas in line with the PHC/UHC agenda in the state. Thus, any individual indicator only gives a partial snapshot of the domain that it represents.

Other attempts to measure equity in PHC services have underscored that addressing social determinants of health and contextual tailored care are key domains [34]. Prior experience and the literature also suggest that non-disease specific and processes of care indicators are important for equitable treatment in PHCs [35]. Our list included indicators related to the environment, contributions of local self-government institutions as well as process indicators related to reporting timeliness.

A major limitation of the study was the fairly narrow group of experts that were involved with the ranking. In as much as we were embedded in ongoing programmatic processes and were also making demands of time from serving officials meant that we had to be conservative and minimise the burden on the system. It was also not feasible to involve a wider range of participants in the ranking, especially those from academia, civil society or even the private sector, although the engagement of such stakeholders was part of the SDG process and continues as the program rolls out. Further, during the ranking exercise, Kerala experienced one of the worst flooding episodes in the state's history. This also created constraints of time and required us to taper the sample for the Delphi. A corollary to this was that the additional indicators recommended by experts while ranking the indicators could not be re-ranked by other experts in a separate round, given time constraints. The scope of this ranking was linked to UHC but focused on PHC; it, therefore, cannot be used for interpretation of broader UHC reform (although our parent study is looking at aspects like service and financial risk protection using secondary sources and primary data collection).

The field testing of the indicators done in three purposively selected facilities in first round was the focus of this paper; later a second round of field testing as well as a detailed household survey were undertaken in systematically chosen additional facilities. Another issue is the heterogeneity of the state and equitable representation of burdens affecting population subgroups–health problems like Kyasanur Forest Disease (KFD), Sickle Cell Anaemia, and Lymphatic Filariasis (LF) disproportionately affect tribal populations but were not selected. As we indicated to facility partners, this kind of context-specific monitoring would be essential moving forward and would have to be developed *in situ* for each district and/or facility based on consensus and local priority-setting.

## Conclusion

In conclusion, this paper describes the process of arriving at and testing the feasibility of a shortlist of indicators to assess primary health care reforms at the facility level in the state of Kerala. Many lessons about indicator development and the health system were learned. The team was given the opportunity to participate in routine state level reviews of the FHC program in which updates regarding the process were presented in front of department officials. In this way, we hoped to ensure alignment and 'interoperability' with national policies (like the National Health Policy 2017) and international priorities like the SDGs, while also being relevant and a starting point to gauge the Aardram reform process in Kerala and addressing the data needs of local managers. These stakeholders provided valuable feedback to practitioners and supervisors on the performance and relevance of the programme. The Delphi provided an opportunity to integrate their experience and expectation into indicator development. Longer term, the inclusion of disaggregated data as well as bespoke indicators based on local priorities can ensure that Kerala is committed to leaving no-one behind.

## Supporting information

**S1 File. Support document on how to rank the indicators in this modified Delphi process.**
(PDF)

**S2 File. Data set of modified Delphi ranking.**
(XLSX)

## Acknowledgments

We are grateful for the important insights and continuous support given to us by the Kerala Department of Health and Family Welfare, the State Health Systems Resource Centre, Kerala (SHSRCK) as well the Aardram Task Force.

## Author Contributions

**Conceptualization:** Devaki Nambiar.

**Data curation:** Hari Sankar D., Jyotsna Negi.

**Formal analysis:** Jyotsna Negi.

**Funding acquisition:** Devaki Nambiar.

**Methodology:** Devaki Nambiar, Hari Sankar D.

**Project administration:** Hari Sankar D., Jyotsna Negi.

**Supervision:** Devaki Nambiar, Arun Nair, Rajeev Sadanandan.

**Validation:** Devaki Nambiar, Jyotsna Negi, Arun Nair, Rajeev Sadanandan.

**Writing – original draft:** Devaki Nambiar, Hari Sankar D.

**Writing – review & editing:** Devaki Nambiar, Hari Sankar D., Jyotsna Negi, Arun Nair, Rajeev Sadanandan.

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
