## [Decision Letter · Decision Letter 0]

23 Apr 2020

PONE-D-20-04694

Monitoring Universal Health Coverage reforms in primary health care facilities: creating a framework, selecting and field-testing indicators in Kerala, India

PLOS ONE

Dear Dr sankar,

Thank you for submitting your manuscript to PLOS ONE. After careful consideration, we feel that it has merit but does not fully meet PLOS ONE’s publication criteria as it currently stands. Therefore, we invite you to submit a revised version of the manuscript that addresses the points raised during the review process.

We would appreciate receiving your revised manuscript by Jun 07 2020 11:59PM. To enhance the reproducibility of your results, we recommend that if applicable you deposit your laboratory protocols in protocols.io, where a protocol can be assigned its own identifier (DOI) such that it can be cited independently in the future. For instructions see: http://journals.plos.org/plosone/s/submission-guidelines#loc-laboratory-protocols

We look forward to receiving your revised manuscript.

Kind regards,

William Joe

Academic Editor

PLOS ONE

Journal Requirements:

2. Thank you for stating the following in the Competing Interests/Financial Disclosure* (delete as necessary) section:

"This work was supported by the Wellcome Trust/DBT India Alliance Fellowship (https://www.indiaalliance.org/ ). Grant number IA/CPHI/16/1/502653) awarded to Dr Devaki Nambiar. The funders had no role in study design, data collection and analysis, decision to publish, or preparation of the manuscript"

We note that one or more of the authors are employed by a commercial company: ACCESS Health.

Reviewers' comments:

Reviewer's Responses to Questions

**Comments to the Author**

1. Is the manuscript technically sound, and do the data support the conclusions?

Reviewer #1: Yes

Reviewer #2: Yes

2. Has the statistical analysis been performed appropriately and rigorously? 

Reviewer #1: Yes

Reviewer #2: Yes

3. Have the authors made all data underlying the findings in their manuscript fully available?

Reviewer #1: Yes

Reviewer #2: Yes

4. Is the manuscript presented in an intelligible fashion and written in standard English?

Reviewer #1: Yes

Reviewer #2: Yes

5. Review Comments to the Author

Reviewer #1: The manuscript is suitable for publication as such. There are no suggestions for improvement from my review. The background is comprehensive with relevant references. The methodology is described in detail involving the various steps in the Delphi method to get feedback from experts. The discussion component is also well written.

Universal health coverage has always been a important concept and the key domain in all WHO initiatives. Hence it is a important manuscript highlighting findings which are of value for policy markers and designing future interventions pertaining to Primary Health Care.

Reviewer #2: Thanks for sending this paper for review to me. I have read this interesting and very policy-relevant piece titled “Monitoring Universal Health Coverage reforms in primary health care facilities: creating a framework, selecting and field-testing indicators in Kerala, India”. Amid COVID-10 outbreak much has been discussed about Kerala Health System, with the help of this review, I got a chance to read about it.

At an outset, I must appreciate the authors for putting together an outstanding piece of work. The work is well conceptualized, methods are standard and the paper is also drafted well. My comments on this piece are minor.

Following important details are missing from the methods section. Adding them will help the readers to judge the robustness of Delphi technique applied in this study.

1. It would have been useful if authors present descriptive statistics (age, gender, occupation/specialisation and years of experience in the particular service) of the participants (or expert panel).

2. How many people were contacted and what is the rate of participation. At one place authors stated that a large group of experts were contacted, but did not mention how many and what are their basic profiles. How did they reduce biased opinions? What is the selection procedure?

3. What kinds of scales are used to assess the agreement on the subjects/items/indicators? It is good if authors present some descriptive statistics of those results.

4. It will also benefit the readers to know at what level consensus or item consolidation was achieved in both the rounds. At what confidence level, authors have decided to include or exclude an indicator/item.

I hope above comments may help the authors.

6. PLOS authors have the option to publish the peer review history of their article (what does this mean?). If published, this will include your full peer review and any attached files.

Reviewer #1: Yes: Dr Muslim Abbas Syed

Reviewer #2: Yes: Srinivas Goli

---

## [Author Response · Author response to Decision Letter 0]

30 May 2020

We thank Reviewer 1 for the comments and support.

Reviewer 2 comments and author responses

1. It would have been useful if authors present descriptive statistics (age, gender, occupation/specialisation and years of experience in the particular service) of the participants (or expert panel).

We have added a Table 2 (see page 14, line 211) and text (see page 13, lines 201-208, underlined in the manuscript) that describe the characteristics of participants. The text is reproduced below

Nearly half of the experts (48%) who participated in two rounds of the exercise were above fifty years of age and almost two-thirds were male (64%) (see Table 2). Most experts had a domain experience of more than ten years (76%) and nearly half of them (48%) were currently serving as state level specialist consultants or program officers (48%) for public health programs. Participants also included those working in a grassroots implementation like field-level health workers, and primary care doctors – almost all experts had experience delivering primary health care at some stage of their careers.

2. How many people were contacted and what is the rate of participation? At one place authors stated that a large group of experts were contacted but did not mention how many and what are their basic profiles. How did they reduce biased opinions? What is the selection procedure?

This text has been added to the manuscript on pages 13 and line number 199-200. It is reproduced below

 A total of 31 participants were invited to participate in two Delphi rounds, out of whom 25 responded (a response rate of 80%, see Figure 2).

The measures to reduce bias in opinion have been added in the manuscript on page 8 line number 169-174 and are reproduced below:

To reduce bias in ranking, the tool consisted of specific information on consideration of how to rank an indicator (S1_File). Experts were encouraged to discuss with their colleagues in the respective domain about the relevance of individual indicators in a comprehensive monitoring framework. A team member visited the experts individually after sending them ranking tool to clarify and doubts in ranking procedure and reiterated the principles to rank them

Information on sampling and the selection procedure has been added on page 7 line number 143-145, It is reproduced below. Profile information, as indicated above is given in table 2 and included in the manuscript on page 14. 

Eligibility criteria for participation were: experience of more than five years in relevant domain; first hand experience with primary care in Kerala; and past or present formal role in design and/or delivery of the FHC program

3. What kinds of scales are used to assess the agreement on the subjects/items/indicators? It is good if authors present some descriptive statistics of those results

Details regarding the scale used are included in the manuscript on page 9 line number 175-181 and added to table 1 on page 10. The text is reproduced below 

Following standard convention and the procedure undertaken in prior ranking exercises [22], mean and median priority scores were calculated for each indicator by creating decision rules based on the distribution of ranks. For the final indicator list, indicators that received a median rank of 1 were included Table 1). Further, Indicators that received a median rank greater than 1 were included only if the mean rank of the indicator was higher than 2.5.

4. It will also benefit the readers to know at what level consensus or item consolidation was achieved in both the rounds. At what confidence level, authors have decided to include or exclude an indicator/item.

As is the convention, the level of consensus was determined quantitatively by obtaining mean and median of ranking of each indicator provided by the experts. We have decided to include or exclude an indicator/item based on the decision rule indicated on page 9, reproduced above.

---

## [Decision Letter · Decision Letter 1]

1 Jul 2020

Monitoring Universal Health Coverage reforms in primary health care facilities: creating a framework, selecting and field-testing indicators in Kerala, India

PONE-D-20-04694R1

Dear Dr. sankar,

We’re pleased to inform you that your manuscript has been judged scientifically suitable for publication and will be formally accepted for publication once it meets all outstanding technical requirements.

Kind regards,

William Joe

Academic Editor

PLOS ONE

Additional Editor Comments (optional):

Reviewers' comments:

Reviewer's Responses to Questions

**Comments to the Author**

1. If the authors have adequately addressed your comments raised in a previous round of review and you feel that this manuscript is now acceptable for publication, you may indicate that here to bypass the “Comments to the Author” section, enter your conflict of interest statement in the “Confidential to Editor” section, and submit your "Accept" recommendation.

Reviewer #1: All comments have been addressed

Reviewer #2: All comments have been addressed

2. Is the manuscript technically sound, and do the data support the conclusions?

Reviewer #1: Yes

Reviewer #2: Yes

3. Has the statistical analysis been performed appropriately and rigorously? 

Reviewer #1: Yes

Reviewer #2: Yes

4. Have the authors made all data underlying the findings in their manuscript fully available?

Reviewer #1: Yes

Reviewer #2: Yes

5. Is the manuscript presented in an intelligible fashion and written in standard English?

Reviewer #1: Yes

Reviewer #2: Yes

6. Review Comments to the Author

Reviewer #1: Thank you for successfully addressing all the points that were mentioned in the previous review. The manuscript is now suitable for publication.

Reviewer #2: Thanks for accepting the suggestion and also providing the additional details. The revised paper is acceptable.

7. PLOS authors have the option to publish the peer review history of their article (what does this mean?). If published, this will include your full peer review and any attached files.

Reviewer #1: No

Reviewer #2: **Yes: **Srinivas Goli

---

## [Editor Report · Acceptance letter]

22 Jul 2020

PONE-D-20-04694R1 

Monitoring Universal Health Coverage reforms in primary health care facilities: creating a framework, selecting and field-testing indicators in Kerala, India 

Dear Dr. Sankar D:

I'm pleased to inform you that your manuscript has been deemed suitable for publication in PLOS ONE. Congratulations! Your manuscript is now with our production department. 

Kind regards, 

on behalf of

Dr. William Joe 

Academic Editor

PLOS ONE